# Contemporary national outcomes of hyperbaric oxygen therapy in necrotizing soft tissue infections

William Toppen[1]*, Nam Yong Cho[2], Sohail Sareh[2], Anders Kjellberg[3,4], Anthony Medak[1], Peyman Benharash[2], Peter Lindholm[1]

1 Division of Hyperbaric Medicine, Department of Emergency Medicine, University of California San Diego, San Diego, CA, United States of America, 2 Cardiovascular Outcomes Research Laboratories, Division of Cardiac Surgery, David Geffen School of Medicine at UCLA, Los Angeles, CA, United States of America, 3 Dept. Physiology and Pharmacology, Karolinska Institute, Stockholm, Sweden, 4 Hyperbaric Medicine, Medical Unit Intensive Care and Thoracic Surgery, Perioperative Medicine and Intensive Care, Karolinska University Hospital, Stockholm, Sweden

* wtoppen@gmail.com

**Data Availability Statement:** All data publicly available from the National Inpatient Sample, HCUP. https://hcup-us.ahrq.gov/nisoverview.jsp.

## Abstract

### Background

The role of hyperbaric oxygen therapy (HBOT) in necrotizing soft tissue infections (NSTI) is mainly based on small retrospective studies. A previous study using the 1998–2009 National Inpatient Sample (NIS) found HBOT to be associated with decreased mortality in NSTI. Given the argument of advancements in critical care, we aimed to investigate the continued role of HBOT in NSTI.

### Methods

The 2012–2020 National Inpatient Sample (NIS) was queried for NSTI admissions who received surgery. 60,481 patients between 2012–2020 were included, 600 (<1%) underwent HBOT. Primary outcome was in-hospital mortality. Secondary outcomes included amputation, hospital length of stay, and costs. A multivariate model was constructed to account for baseline differences in groups.

### Results

Age, gender, and comorbidities were similar between the two groups. On bivariate comparison, the HBOT group had lower mortality rate (<2% vs 5.9%, p<0.001) and lower amputation rate (11.8% vs 18.3%, p<0.001) however, longer lengths of stay (16.9 days vs 14.6 days, p<0.001) and higher costs ($54,000 vs $46,000, p<0.001). After multivariate analysis, HBOT was associated with decreased mortality (Adjusted Odds Ratio (AOR) 0.22, 95% CI 0.09–0.53, P<0.001) and lower risk of amputation (AOR 0.73, 95% CI 0.55–0.96, P = 0.03). HBO was associated with longer stays by 1.6 days (95% CI 0.4–2.7 days) and increased costs by $7,800 (95% CI $2,200-$13,300), they also had significantly lower risks of non-home discharges (AOR 0.79, 95%CI 0.65–0.96).

**Funding:** The author(s) received no specific funding for this work.

**Competing interests:** The authors have declared that no competing interests exist.

## Conclusions

After correction for differences, HBOT was associated with decreased mortality, amputations, and non-home discharges in NSTI with the tradeoff of increase to costs and length of stay.

## Background

Necrotizing soft tissue infection (NSTI) is a debilitating and potentially life-threatening condition that affects the skin, underlying tissue, muscle, and fascia. While necrotizing fasciitis and gas gangrene are separate entities, their clinical managements are similar [1]. **No high grade evidence is available for any of the interventions in NSTI [2].** International consensus statements on the most significant factors for reducing mortality including early diagnosis, operative debridement, and broad-spectrum antibiotics exists, but adjunct therapies such as hyperbaric oxygen therapy (HBOT) remain controversial [3–5]. While HBOT has been used for nearly half a century to supplement traditional treatments for NSTI, high-level clinical evidence has yet to be available [6]. Current recommendations for HBOT are based on limited prospective and retrospective cohort studies [4]. A previous analysis of 45,913 patients in the National Inpatient Sample (NIS) from 1988 to 2009 showed a statistically significant reduction in mortality albeit with higher hospitalization costs and longer length of stay [7]. A retrospective analysis of a Danish cohort with 1527 patients between 2005–2018 reported that, despite increasing incidence and high mortality overall, HBOT was associated with decreased 30- and 90-day mortality in high volume centers [8]. A recent systematic review and meta-analysis including 49,152 patients (1448 received HBOT) from 23 non-randomized studies spanning from 1990–2022 indicated that patients treated with HBOT for NSTI had reduced risk of mortality and incidence of complications, RR 0.52 (95% CI 0.40–0.68, p = 0.03) [9]. Given the argument of updated guidelines in critical care [10–12], we aimed to investigate the continued role of HBOT for patients with NSTI and a subgroup analysis of sepsis from 2012 onward. Our primary question was whether HBOT would still demonstrate beneficial results using data from the latest update to the NIS.

## Methods

### Data source and study cohort

This was a retrospective cohort study of the 2012–2020 National Inpatient Sample (NIS). As part of the Healthcare Cost and Utilization Project (HCUP), the NIS is the largest all-payer inpatient database in the United States and provides accurate estimates for greater than 95% of all hospitalizations. Weighted results are statistically extrapolated from the sample to estimate national totals while the unweighted data is the raw patient data compiled from 20% of the admissions at participating sites. For this study we present unweighted data only. Of note, Healthcare Cost and Utilization Project (HCUP) privacy protection requirements do not allow the reporting of data where there are less than or equal to 10 individuals total in the NIS database. Database was first accessed for this project December 7$^{th}$, 2022. Researchers did not have access to individually identifying data during or after completion of the study as it was not included in the database.

All adult (age ≥ 18 years) hospitalizations entailing surgical interventions with a diagnosis of NSTI were identified using relevant International Classification of Diseases, Ninth or Tenth

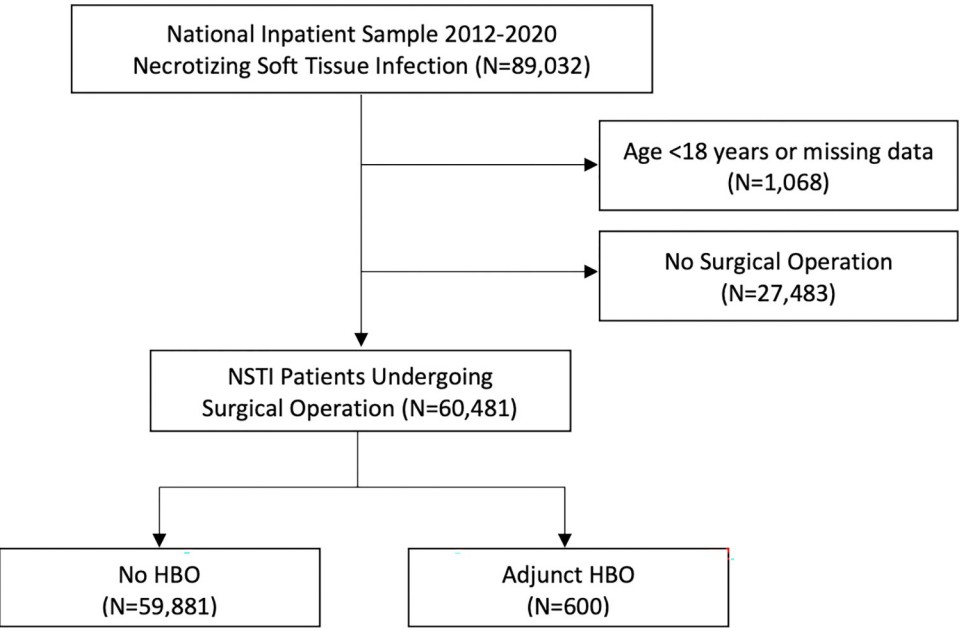

**Fig 1. Flow diagram of patients hospitalized for necrotizing soft tissue infection stratified by hyperbaric oxygen therapy.**

Revision (ICD-9/10) codes (S1 Table-Codes). Patients were stratified into those who received hyperbaric oxygen therapy (HBOT) and those who did not. Of note, available data did not include HBOT duration, number of treatments, treatment pressure, or timing of treatments in relation to time of admission or surgery. Patients who did not undergo surgical interventions or those with missing key variables, including age, mortality, and costs, were excluded from the study (1.2%). (Fig 1).

## Study variables and outcomes

Patient and hospital characteristics such as age, gender, hospital region, income quartile, and insurance status were defined according to the NIS data dictionary [13]. The burden of chronic illness on the cohort was assessed via the Van Walraven modification of the Elixhauser Comorbidity Index (ECI) [14]. In addition, specific patient data, including NSTI location and type of pathogen, was defined as previously in the literature [7]. The primary outcome of interest was in-hospital mortality, while secondary outcomes included index hospitalization length of stay (LOS), costs, amputation, and non-home discharge. Non-home discharge was defined as a transfer to short-term care or a skilled nursing facility.

## Statistical methods

Categorical variables are reported as a percentage and continuous variables as mean with a 95% Confidence Interval (CI). The chi-square and the Student's T-tests were used to analyze patient characteristics in both study cohorts. Nonparametric rank-based tests were used to assess temporal trends. Hospitalization costs were calculated by applying center-specific cost-to-charge ratios to total hospitalization charges and adjusting for inflation using the 2020 Personal Health Index. Multivariable regression was used to evaluate patient and hospital factors associated with in-hospital mortality as well as secondary outcomes. Covariate analysis for this model was assisted by Elastic Net regularization to improve out-of-sample generalizability.

S3 Table lists all included covariates. Regression outputs are reported as adjusted odds ratios (AOR) or beta coefficients (β) with 95% confidence intervals (95% CI). A P-value <0.05 was considered statistically significant. All statistical analyses were performed using Stata 16.1 (StataCorp, College Station, TX).

### Subgroup analyses

Several subgroup analyses were also completed using the methods outlined above. In addition to the primary analysis, we performed the same analysis on a subset of all patients who also presented with sepsis as coded in the NIS. Additionally, separate analyses were done on patients as subdivided by sites of NSTI (truncal versus extremity) and causative pathogen (Clostridial versus non-Clostridial).

### Ethics

Our Institutional Review Board (IRB) granted an exemption for this analysis of retrospective data, #806108, Dec 08, 2022. Patient consent was waived by the IRB.

## Results

### Demographics

Of 60,481 unweighted patients admitted with a diagnosis of NSTI who underwent surgery, 600 (<1%) received HBOT during their admission. As shown in Table 1, both groups had similar ages, gender distribution, the burden of comorbidities measured by ECI, incidence of AKI, and causative pathogens. Overall, patients in the HBOT group had a lower incidence of sepsis (49.2 vs 59.4%, P<0.001) and "urgent" admissions (90.7 vs 93.9%, P<0.001) compared to those in the control group. In addition, patients in the HBOT cohort had a higher incidence of truncal NSTI (17.5% vs 2.6%, P<0.001) and more private insurance (34.5 vs 25.2%, P<0.001), compared to others. Patients in the HBOT group were more likely to be from the Midwestern United States than those in the control (30.3 vs 20.0%), while those in the control group were more likely to be from the Western United States (21.3 vs 10.2%, P = 0.01). Patients in the HBOT group were more frequently from the highest income quartile (18.0 vs 15.8%), while the control group had a higher proportion of those in the lowest income quartile (37.5 vs 30.5%, P = 0.04). (Table 1). As shown in Table 1, patients undergoing HBOT for NSTI had greater rates of CKD (24.2 vs 19.5%, p = 0.004), diabetes (70.8 vs 61.4%, p<0.001) and peripheral arterial disease (13.0 vs 8.2%, p<0.001).

### Unadjusted outcomes

On bivariate comparison, the control group had a higher incidence of mortality (5.9 vs <2%, P<0.001) and amputations (18.3% vs 11.8%, P<0.001). The HBOT group had significantly longer hospital stays (17 days vs 15 days, P<0.001) and higher costs ($54,000 vs $46,000 (P<0.001). Rates of non-home discharge (discharge other than home or home with home health) were higher in the control group (45.3 vs 41%, P<0.001). (Table 2)

### Risk-adjusted outcomes

After adjusting for baseline differences between groups as previously described, HBOT was associated with a lower risk of mortality (AOR 0.22, 95% CI 0.09–0.53, P<0.001) and lower risk of amputation (AOR 0.73, 95% CI 0.55–0.96, P = 0.03). Management of NSTI with HBOT was shown to be associated with longer hospital stays by 1.6 days (95%CI 0.4–2.7 days) and increased costs by $7,800 (95%CI $2,200-$13,300). In addition, HBOT status was associated

**Table 1. Demographics.**

| | HBOT (n = 600) | Control (n = 59,881) | P-Value |
|---|---|---|---|
| Mean Age, year (95% CI) | 55.4 (53.8–57.0) | 55.2 (55.0–55.3) | 0.58 |
| Gender, (%) | | | 0.30 |
| Male | 394 (65.7) | 38,690 (65.6) | |
| Female | 206 (34.3) | 21,173 (34.4) | |
| Hospital Bed Size, (%) | | | 0.99 |
| Small | 90 (15.0) | 9,570 (16.5) | |
| Medium | 130 (21.7) | 17,291 (29.3) | |
| Large | 380 (63.3) | 33,020 (54.2) | |
| Hospital Location, (%) | | | 0.01 |
| Northeast | 77 (12.8) | 10,093 (16.5) | |
| Midwest | 182 (30.3) | 11,984 (20.0) | |
| South | 280 (16.7) | 25,171 (42.2) | |
| West | 61 (10.2) | 12,633 (21.3) | |
| Location/teaching status, (%) | | | 0.80 |
| Rural | 25 (4.2) | 4,340 (7.0) | |
| Urban non-teaching | 87 (14.5) | 11,440 (17.3) | |
| Urban teaching | 488 (81.3) | 44,101 (75.7) | |
| Income Quartile, (%) | | | <0.001 |
| First (Lowest) | 183 (30.5) | 22,473 (37.5) | |
| Second | 176 (29.3) | 15,635 (26.1) | |
| Third | 132 (22.0) | 12,316 (20.6) | |
| Fourth (Highest) | 110 (18.3) | 9,461 (15.8) | |
| Insurance Status, (%) | | | <0.001 |
| Medicare | 220 (36.7) | 22,634 (37.8) | |
| Medicaid | 110 (18.3) | 14,405 (24.1) | |
| Private Insurance | 207 (34.5) | 15,113 (25.2) | |
| Self-pay | 45 (7.5) | 5,134 (8.6) | |
| Other | 18 (3.0) | 2,575 (4.3) | |
| Elixhauser Comorbidity Index (95% CI) | 3.4 (3.3–3.6) | 3.4 (3.4–3.4) | 0.86 |
| Acute Kidney Injury, (%) | 250 (41.7) | 25,001 (41.8) | 0.48 |
| Cardiac Arrest, (%) | ≤10 | 1,262 (2.1) | 0.06 |
| **Chronic Kidney Disease, (%)** | 145 (24.2) | 11,696 (19.5) | 0.004 |
| **Diabetes, (%)** | 425 (70.8) | 36,743 (61.4) | <0.001 |
| Sepsis, (%) | 295 (49.2) | 34,968 (59.4) | <0.001 |
| **Peripheral Arterial Disease, (%)** | 78 (13.0) | 4,892 (8.2) | <0.001 |
| Pneumonia, (%) | 21 (3.5) | 3,557 (5.9) | 0.99 |
| Pulmonary Embolism, (%) | ≤10 | 597 (1.0) | 0.11 |
| Admission Type | | | <0.001 |
| Urgent | 544 (90.7) | 56,224 (93.9) | |
| Elective | 56 (9.3) | 3,657 (6.1) | |
| Site of NSTI | | | <0.001 |
| Truncal | 105 (17.5) | 1,582 (2.6) | |
| Extremity | 495 (82.5) | 58,299 (97.4) | |
| Pathogen of NSTI | | | 0.01 |
| Clostridial | 109 (18.2) | 13,591 (22.7) | |

*(Continued)*

**Table 1.** (Continued)

| | HBOT (n = 600) | Control (n = 59,881) | P-Value |
|---|---|---|---|
| Non-Clostridial | 491 (81.8) | 46,290 (77.3) | |

**Table 1.** Demographics and baseline characteristics of the hyperbaric oxygen therapy (HBOT) and non-HBOT therapy group for patients undergoing surgical intervention for necrotizing soft tissue infection. Data is presented as mean (95% CI), or number (%). Abbreviations: NSTI, necrotizing soft tissue infection. CI, Confidence Interval.

with lower odds of non-home discharge, with a lower risk of discharge to a short-term care facility (AOR 0.47, 95% CI 0.6–0.95, P = 0.008) (Table 3).

## Subgroup analyses

**Sepsis cohort.** A subgroup analysis was done to include only patients who additionally presented with sepsis as coded in the NIS. Total number of patients was 35,258, of which 295 (<1%) received HBOT. Univariate comparison of sepsis-only subgroups is shown in S2 Table. After correcting for differences between the groups using the same multivariate regression model, HBOT continued to be associated with a significantly lower mortality rate (AOR 0.16, 95% CI 0.06–0.46), lower odds of amputations (AOR 0.42, 95% CI 0.28–0.62), and fewer non-home discharges (AOR 0.73, 95%CI 0.56–0.95). In contrast to the primary analysis, after adjustments, neither length of stay nor cost was significantly different between groups (Table 4).

**Location of NSTI.** On univariate analysis, HBOT was associated with decreased mortality in both truncal and extremity NSTI (Table 5). After adjusting for relevant confounders, this difference persisted in the group with NSTI of the extremity (AOR 0.19, 95% CI 0.08–0.50, P<0.001).

**Pathogen.** There was insufficient incidence of deaths with confirmed Clostridial infections to perform a meaningful multivariate analysis. When looking at non-Clostridial infections, the unadjusted analysis found an association between HBOT and decreased mortality (OR 0.17, P<0.001). After adjusting for relevant confounders, this difference persisted (AOR 0.22, 95% CI 0.09–0.52), P<0.001). (Table 5).

**Table 2. Bivariate outcomes.**

| | HBOT (n = 600) | Control (n = 59,881) | P-Value |
|---|---|---|---|
| Mortality, (%) | ≤10 (<0.02%) | 3,505 (5.9) | <0.001 |
| Amputations, (%) | 71 (11.8) | 10,957 (18.3) | <0.001 |
| Length of Stay, (Days) | 16.9 (15.2–18.5) | 14.6 (14.4–14.8) | <0.001 |
| Cost, ($1,000) | 53.6 (48.4–58.9) | 46.0 (45.1–46.8) | <0.001 |
| Discharge Disposition, (%) | | | <0.001 |
| Home | 166 (27.7) | 13,560 (22.6) | |
| Short-term Hospital | 14 (2.3) | 3,146 (5.3) | |
| Skilled Nursing Facility | 232 (38.7) | 23,976 (40.0) | |
| Home Health Care | 166 (27.7) | 14,781 (24.7) | |
| Against Medical Advice | 15 (2.5) | 910 (1.5) | |
| Non-home Discharge, (%) | 246 (41.0) | 27,122 (45.3) | <0.001 |

**Table 2.** Results of bivariate analysis comparing outcomes in the hyperbaric oxygen therapy (HBOT) and non-HBOT therapy group in patients undergoing surgical intervention for necrotizing soft tissue infection. AKI, Acute Kidney Injury. Continuous variables are reported as mean (95% Confidence Interval), or number (%).

**Table 3. Risk-adjusted outcomes.**

|  | Estimates | 95% CI | P-Value |
|---|---|---|---|
| Mortality | 0.22* | 0.09–0.53 | <0.001 |
| Amputation | 0.73* | 0.55–0.96 | 0.03 |
| LOS, (Days) | 1.58* | 0.43–2.74 | 0.007 |
| Cost, ($1,000) | 7.8* | 2.2–13.3 | 0.006 |
| Discharge Disposition (REF: Routine) |  |  |  |
| Short-term Hospital | 0.47* | 0.27–0.82 | 0.008 |
| Skilled Nursing Facility | 0.75* | 0.60–0.95 | 0.02 |
| Home Health Care | 0.80 | 0.64–1.01 | 0.07 |
| Against Medical Advice | 1.60 | 0.92–2.77 | 0.09 |
| Non-home Discharge | 0.79 | 0.65–0.96 | 0.02 |

**Table 3.** Risk-adjusted outcomes in patients undergoing surgical intervention for necrotizing soft tissue infection with concomitant hyperbaric oxygen therapy (HBOT) (Reference: non-HBOT therapy). Estimates are reported as AORs or β-coefficients for binary and continuous variables, respectively.

## Discussion

Despite improvements in critical and surgical care over the last decade, our data shows that hyperbaric oxygen continues to be associated with significantly increased survivability in necrotizing soft tissue infections. After adjustments for confounders, the addition of HBOT to surgery was associated with decreased mortality, decreased amputation rates, and more discharges to home. However, our analysis revealed the apparent tradeoff of HBOT being associated with longer total lengths of stay and slightly higher hospital costs. In the subgroup of patients with sepsis, decreases in mortality, amputations, and non-home discharges persisted, however without the increases in length of stay and cost. One decade prior to our study, Soh et al investigated an earlier version of the NIS and similarly reported a significant reduction in mortality associated with HBO in NSTI (OR 0.49, 95%-CI 0.29–0.83). More recently Hedetoft et al's work with the Danish national registry reported improvements in 30-day mortality (OR 0.54, 95% CI 0.33 to 0.91, p = 0.02) and 90-day mortality (OR 0.61, 95% CI 0.39 to 0.97, p = 0.03) [8] Despite these findings and many other studies detailing significant improvements

**Table 4. Sepsis-only outcomes.**

|  | Estimates | 95% CI | P-Value |
|---|---|---|---|
| Mortality | 0.16* | 0.06–0.46 | <0.001 |
| Amputation | 0.42* | 0.28–0.62 | <0.001 |
| LOS, (Days) | -0.34 | -2.21–1.53 | 0.72 |
| Cost, ($1,000) | 3.4 | -5.6–12.3 | 0.46 |
| Discharge Disposition (REF: Routine) |  |  |  |
| Short-term Hospital | 0.44* | 0.22–0.89 | 0.02 |
| Skilled Nursing Facility | 0.64* | 0.46–0.90 | 0.01 |
| Home Health Care | 0.78 | 0.55–1.10 | 0.16 |
| Against Medical Advice | 1.46 | 0.62–3.41 | 0.39 |
| Non-home Discharge | 0.73* | 0.56–0.95 | 0.02 |

**Table 4.** Risk-adjusted outcomes in patients with sepsis undergoing surgical intervention for necrotizing soft tissue infection with concomitant hyperbaric oxygen (HBOT) therapy (Reference: non-HBOT therapy). Estimates are reported as AORs or β-coefficients for binary and continuous variables, respectively.

**Table 5. Subgroup analysis.**

| | Mortality with HBOT (n, %) | Mortality without HBOT (n, %) | Unadjusted OR for death (%) | P-Value | Adjusted OR for death (95% CI) | P-Value |
|---|---|---|---|---|---|---|
| **Site of NSTI** | | | | | | |
| Truncal[†] | ≤ 10 (≤ 0.8%) | 55 (1.8) | 0.94 | <0.001 | 1.19 (0.06–2.04) | 0.91 |
| Extremity[†] | ≤ 10 (≤ 0.8%) | 7,800 (5.8) | 0.13 | <0.001 | 0.19 (0.08–0.50) | <0.001 |
| **Pathogen of NSTI** | | | | | | |
| Clostridial [†] | ≤ 10 (≤ 0.8%) | 660 (2.1) | - | - | - | - |
| Non-Clostridial | 15 (1.2) | 7,195 (6.8) | 0.17 | <0.001 | 0.22 (0.09–0.52) | <0.001 |
| | **Complications with HBOT (n, %)** | **Complications without HBOT (n, %)** | **Unadjusted OR for complications (%)** | **P-Value** | **Adjusted OR for complications (95%CI)** | **P-Value** |
| **Site of NSTI** | | | | | | |
| Truncal | 225 (75.0) | 2,335 (75.0) | 0.99 | <0.001 | 1.14 (0.58–2.23) | 0.71 |
| Extremity | 1,060 (82.2) | 110,115 (82.4) | 0.99 | <0.001 | 1.03 (0.74–1.44) | 0.84 |
| **Pathogen of NSTI** | | | | | | |
| Clostridial | 245 (81.7) | 24,930 (80.8) | 1.01 | <0.001 | 2.45 (1.33–4.48) | 0.004 |
| Non-clostridial | 1,040 (80.6) | 87,519 (82.6) | 0.97 | <0.001 | 0.74 (0.52–1.06) | 0.10 |
| | **Amputation with HBOT (n, %)** | **Amputation without HBOT (n, %)** | **Unadjusted OR for Amputation (%)** | **P-Value** | **Adjusted OR for Amputation (95%CI)** | **P-Value** |
| Extremity NSTI | 185 (14.3) | 25,440 (19.0) | 0.75 | <0.001 | 0.73 | 0.03 |

**Table 5.** Unadjusted and adjusted outcomes in patients with NSTI undergoing surgical intervention for necrotizing soft tissue infection with concomitant HBOT presented as mean (95% CI), or number (%). Adjusted odds ratio (OR) for death and complications with hyperbaric oxygen therapy (HBOT) after stratification (Reference: non-HBOT cohort). Complications included cardiovascular (arrest, tamponade, arrhythmia), respiratory (failure, pneumonia, prolonged ventilation) and infectious (sepsis surgical site infection) etiologies.

[†]HCUP privacy protection requirements do not allow the reporting of data where there are less than or equal to 10 individuals records in a given cell. Abbreviations: NSTI, Necrotizing Soft-Tissue Infection. CI, Confidence Interval.

in patient outcomes with the addition of HBOT to NSTI, utilization rates remain remarkably low nation-wide.

There are several proposed and well-studied mechanisms by which HBOT may be effective adjective therapy in the management of NSTI. Traditionally HBOT was advocated and used mainly for clostridial infections (gas gangrene) since high oxygen tension in the tissues creates an inhospitable environment for the obligate anaerobes involved in the deadliest form of NSTI [15, 16]. In the case of clostridial infections, there is persuasive evidence that hyperbaric oxygen therapy at sufficiently high partial pressures halts the production of alpha-toxin, buying precious time for antibiotics to take effect [17]. Modern understanding of host-pathogen interaction has advanced the use of HBOT beyond gas gangrene to various complicated chronic and acute infections [18]. Tissue hypoxia below 30mmHg impedes granulocyte and macrophage function, a situation commonly seen in NSTI as infection progresses [19]. Hyperbaric oxygen can assist by intermittently increasing the oxygen diffusion gradient, enabling normal function of the innate immune system and antibiotics as well as improving survival of threatened cells [20–22]. Despite improvements in the understanding and management of NSTI, mortality is still very high and survivors are left with debilitating amputations, tissue defects, and scars [23].

Within the subgroup of patients with NSTI and sepsis, the association with HBOT and decreased mortality, amputations, and non-home discharges persisted while the signals for both longer length of stay and increased costs were no longer seen. Several iterations of work

on the role of HBOT in sepsis has found that in mice models, survivability is significantly increased with early HBOT. The proposed mechanism involves the ability of HBOT to down-regulate inflammatory cytokines such as TNF-alpha, IL-6, and modulate IL-10 leading to a reduction of the systemic inflammatory response. Indeed, cultured macrophages similarly show a decreased cytokine release when exposed to hyperbaric oxygen conditions. [24–26]. These findings strengthen the argument for the utilization of HBOT early in the disease course, especially in the most severely ill patients.

The increased cost associated with HBOT should be weighed against the financial burden of mortality and morbidity. During the Sars-COV2 (COVID19) pandemic and the resultant strain on medical resources, a renewed interest was placed on appropriate resource allocation and re-evaluating the value of a statistical life (VSL) through the lens of finite medical resources. While the VSL is not a perfect corollary as its main utility is comparison of wage paid versus occupation risk, it is nevertheless a useful benchmark when comparing, in broad strokes, the financial viability of a potentially lifesaving treatment. For the purposes of risk analysis, the United States government values a statistical life at $7.4 million USD [27]. By this framework, an intervention that reduces risk by half, for instance, would be considered cost-effective if it cost less than $3.7 million dollars. In addition to mortality, amputation is a common complication of NSTI leading to significant short- and long-term morbidity for patients. Gomez et al looked at the overall cost of amputations from workplace accidents and reported that, depending on anatomy of amputation and number of amputations, costs may be as high as $46,000 and result in several months of lost work days [28]. Our study found an association with significant decrease in the odds of both mortality and amputation among NSTI patients treated with HBOT. Though our study found an estimated increase in costs of $7,800 in patients who received HBOT, and while a detailed financial analysis is outside the scope of this publication, the relatively small difference in hospital costs for a significant reduction in death and amputations make a strong case for the cost-effectiveness of HBOT in NSTI.

Our study does have several limitations inherent to its nature as a large retrospective review of a national database. Temporal relations within specific hospitalizations are difficult to ascertain, for example timing of HBOT from presentation, timing of surgery, etc. Data was not available with regards to HBO treatment protocol (duration, treatment depth, number of treatments completed) nor whether chambers were multiplace or monoplace. In fact, this shortcoming may have actually served to dampen the positive results of HBOT, as there may be patients with only one treatment or delayed HBOT included in the treatment group. With the retrospective nature of this work, it is impossible to guarantee standardization of any treatment modality (HBOT, surgery, antibiotics, etc) between patients. Sepsis diagnosis is based on ICD-10 as registered; the register does not contain any information on adherence to current sepsis guidelines, but the overall low mortality rate suggests that most patients did not have septic shock [10]. From experience and a previous prospective cohort study, we know that there is an inherent selection bias for the administration of HBOT depending on multiple factors such as availability of ICU resources and hyperbaric chambers at the receiving hospital, and hemodynamic stability of the patient [29]. Our data does not include information on disease severity, such as sequential organ failure assessment score (SOFA) or hemodynamic parameters. Patient groups were heterogeneous in several aspects, though best efforts were made to account for this through statistical normalization and modeling. Lastly, complications of HBOT can be vague, nebulous, and shared with several diseases or iatrogenic processes making it difficult and ill-advised to draw useful conclusions regarding their rate or severity from such a large database.

Our study has several strengths worth highlighting. The NIS allows us to amass a cross-section of real patients from across the country to evaluate real-world application of HBOT in NSTI rather than animal models or restrictive homogenous patient populations. Our robust

multivariate model allows adjustment for numerous potential confounders between our two patient groups. Lastly, we follow in the footsteps of Soh et al. [7] and by employing the same large, nationwide database of hospital admission for NSTI we are able to evaluate the association between HBOT and NSTI through time and advances in surgical and critical care.

Necrotizing soft tissue infections can be rapidly progressive, life threatening, and often leave patients with chronic disfigurements. While the medical field agrees regarding the critical role of prompt antibiotics and surgical debridement, disagreement as to the role of HBOT remains pervasive [3, 4]. Our study represents the largest single study of the role of HBOT in NSTI to date. We found that after adjusting for relevant confounders, HBOT added to surgery and antibiotics was associated with decreased risk of mortality, amputations, and non-home discharges with the trade-off of slightly increased costs and longer lengths of stay. In a subset of patients with NSTI and sepsis however, improved rates of mortality, amputations, and non-home discharges persisted, without the signal for increased costs or lengths of stay. To gain a more comprehensive understanding of the role of HBOT in NSTI, with the ultimate aim of improving patient outcomes and changing evidence-based clinical practice, a well-designed randomized controlled, multicenter trial is warranted.

## Supporting information

**S1 Table. ICD-10 codes included in data retrieval from National Inpatient Sample database.**
(DOCX)

**S2 Table. Results of bivariate analysis comparing outcomes in the hyperbaric oxygen therapy (HBOT) and non-HBOT therapy group in patients with sepsis undergoing surgical intervention for necrotizing soft tissue infection.** AKI, Acute Kidney Injury. Continuous variables are reported in mean (95% Confidence Interval).
(DOCX)

**S3 Table. List of covariates included in multivariate analysis.**
(DOCX)

## Author Contributions

**Conceptualization:** William Toppen, Nam Yong Cho, Sohail Sareh, Anthony Medak, Peyman Benharash, Peter Lindholm.

**Data curation:** William Toppen, Nam Yong Cho.

**Formal analysis:** William Toppen, Nam Yong Cho, Sohail Sareh, Anders Kjellberg, Peyman Benharash, Peter Lindholm.

**Investigation:** William Toppen, Nam Yong Cho.

**Methodology:** William Toppen, Sohail Sareh, Anders Kjellberg, Peter Lindholm.

**Project administration:** William Toppen, Anthony Medak, Peyman Benharash.

**Software:** Sohail Sareh.

**Supervision:** Sohail Sareh, Peyman Benharash, Peter Lindholm.

**Validation:** Sohail Sareh.

**Writing – original draft:** William Toppen, Nam Yong Cho, Sohail Sareh, Anders Kjellberg, Anthony Medak.

**Writing – review & editing:** William Toppen, Nam Yong Cho, Sohail Sareh, Anders Kjellberg, Anthony Medak, Peyman Benharash, Peter Lindholm.

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
