## [Decision Letter · Decision Letter 0]

25 Jan 2024

PONE-D-23-35133Contemporary National Outcomes of Hyperbaric Oxygen Therapy in Necrotizing Soft Tissue InfectionsPLOS ONE

Dear Dr. Toppen,

Thank you for submitting your manuscript to PLOS ONE. After careful consideration, we feel that it has merit but does not fully meet PLOS ONE’s publication criteria as it currently stands. Therefore, we invite you to submit a revised version of the manuscript that addresses the points raised during the review process.

Please see the comments below from five reviewers. Most reviewers have raised only a few concerns, and you are welcome to cross-reference your responses to their concerns, if any are repeated, and you may rebut any comments that you feel are outside the scope of the study. Please ensure that the statistical analysis is presented and fully reproducible by another researcher.

We look forward to receiving your revised manuscript.

Kind regards,

Hanna Landenmark

Staff Editor

PLOS ONE

Journal Requirements:

Reviewers' comments:

Reviewer's Responses to Questions

**Comments to the Author**

1. Is the manuscript technically sound, and do the data support the conclusions?

Reviewer #1: Partly

Reviewer #2: No

Reviewer #3: Yes

Reviewer #4: Yes

Reviewer #5: Yes

2. Has the statistical analysis been performed appropriately and rigorously? 

Reviewer #1: No

Reviewer #2: No

Reviewer #3: I Don't Know

Reviewer #4: Yes

Reviewer #5: Yes

3. Have the authors made all data underlying the findings in their manuscript fully available?

Reviewer #1: Yes

Reviewer #2: No

Reviewer #3: Yes

Reviewer #4: Yes

Reviewer #5: Yes

4. Is the manuscript presented in an intelligible fashion and written in standard English?

Reviewer #1: Yes

Reviewer #2: Yes

Reviewer #3: Yes

Reviewer #4: Yes

Reviewer #5: Yes

5. Review Comments to the Author

Reviewer #1: Dear Authors,

I appreciate the opportunity to review your manuscript " Contemporary National Outcomes of Hyperbaric Oxygen Therapy in Necrotizing Tissue Infections."

I have several comments/questions for your consideration:

- Given the complex survey design of the the NIS data why is it that the authors have chosen not to apply the survey design features of the NIS to obtain weighted estimates?

- Please clearly state what variables were included in the adjusted model.

- Consider the application of a statistical approach to account for potential allocation bias for the hyperbaric oxygen therapy.

- Consider including specific comorbid medical conditions ( i.e. Diabetes, CKD, Peripheral artery disease, etc.) in addition to the Elixhauser Comorbidity Index.

- Is hyperbaric oxygen therapy more common to be an inpatient versus an outpatient procedure?

Minor comments

- Review Table 3 and Table 4 under discharge disposition. Currently written " Against Medical DEVICE" instead of "Against Medical Advice"

- line 325: citation method needs to be consistent

Reviewer #2: Rejected with major improvements in the content.

The topic of the article has been repetitively studied in many articles and there is no mew information which is being revealed through this study and it does not seem to have any impactful findings

Reviewer #3: The article deals with a very interesting topic, about which there are not many articles covered in this way, which gives the article importance. The topic is presented in an appropriate way and, in my opinion, will contribute to the scientific community in the best possible way.

I am of the opinion that, after checking the statistical data, the paper can be published

Reviewer #4: The authors analysis on utility of Hyperbaric oxygen therapy is a valuable approach. The emphasis should be more on making availability of multiple center of Hyperbaric oxygen therapy and making treatment cheaper rather than concentration on the cost incurred for the treatment. As the author could not collect the data on HBOT duration, number of treatments, treatment pressure, or timing of treatments in relation to time of admission or surgery which are critical data necessary for the management of such patients, the international standards of application of HBOT could be discussed in discussion part for enlightening the reader's knowledge. The Author should get the data of other indications used for Hbot and discuss with the further potentials of HBOT in clinical practice and future publications.

Reviewer #5: I like to thank the authors for preparing this manuscript. In this manuscript entitled “Contemporary National Outcomes of Hyperbaric Oxygen Therapy in Necrotizing Soft Tissue Infections”, they present a carefully conducted analysis of a large data pool regarding the in-hospital outcomes of patients admitted with NSTI who received hyperbaric oxygen therapy. The authors have described the results in a very comprehensive manner. However, I have few comments as below:

1. In table 2, to be more explicit, please include % (in addition to <10) of mortality in HBOT column. In cost row and control column, please remove 385 number if it has been added there by mistake.

2. In table 5, to be mor explicit, please include % (in addition to <10 number) in mortality rows.

6. PLOS authors have the option to publish the peer review history of their article (what does this mean?). If published, this will include your full peer review and any attached files.

Reviewer #1: No

Reviewer #2: No

Reviewer #3: No

Reviewer #4: No

Reviewer #5: No

---

## [Author Response · Author response to Decision Letter 0]

27 Feb 2024

We extend our deepest gratitude to the reviewers for their time and consideration of our manuscript. We have addressed each of their concerns and questions individually in the attached "Response to Reviewers" file. We hope that this answers are felt to be satisfactory and appreciate the opportunity to strengthen our work.

---

## [Decision Letter · Decision Letter 1]

5 Mar 2024

Contemporary National Outcomes of Hyperbaric Oxygen Therapy in Necrotizing Soft Tissue Infections

PONE-D-23-35133R1

Dear Dr. Toppen,

We’re pleased to inform you that your manuscript has been judged scientifically suitable for publication and will be formally accepted for publication once it meets all outstanding technical requirements.

Kind regards,

Andrea Martinuzzi

Academic Editor

PLOS ONE

Additional Editor Comments (optional):

Reviewers' comments:

Reviewer's Responses to Questions

**Comments to the Author**

1. If the authors have adequately addressed your comments raised in a previous round of review and you feel that this manuscript is now acceptable for publication, you may indicate that here to bypass the “Comments to the Author” section, enter your conflict of interest statement in the “Confidential to Editor” section, and submit your "Accept" recommendation.

Reviewer #3: All comments have been addressed

Reviewer #4: All comments have been addressed

2. Is the manuscript technically sound, and do the data support the conclusions?

Reviewer #3: Yes

Reviewer #4: Yes

3. Has the statistical analysis been performed appropriately and rigorously? 

Reviewer #3: Yes

Reviewer #4: Yes

4. Have the authors made all data underlying the findings in their manuscript fully available?

Reviewer #3: Yes

Reviewer #4: Yes

5. Is the manuscript presented in an intelligible fashion and written in standard English?

Reviewer #3: Yes

Reviewer #4: Yes

6. Review Comments to the Author

Reviewer #3: Always a current topic of NF, considering today's abuse of antibiotics and increasing resistance of microorganisms.

The work is well conceived, it contains all the relevant parts

The abstract is well-conceived, short and clear.

In the introduction, it is necessary to provide adequate literary citations for the presented data.

Results are presented clearly

In the discussion, the authors listed all the relevant data that support the presented study.

The conclusion follows from the displayed text.

Reviewer #4: Future prospective studies are required for proper evaluation. However this initial manuscript will help the reader to know the application of HBO in NSTI patients

7. PLOS authors have the option to publish the peer review history of their article (what does this mean?). If published, this will include your full peer review and any attached files.

Reviewer #3: No

Reviewer #4: No

---

## [Editor Report · Acceptance letter]

12 Mar 2024

PONE-D-23-35133R1 

PLOS ONE

Dear Dr. Toppen, 

I'm pleased to inform you that your manuscript has been deemed suitable for publication in PLOS ONE. Congratulations! Your manuscript is now being handed over to our production team.

Kind regards, 

on behalf of

Dr. Andrea Martinuzzi 

Academic Editor

PLOS ONE